# Tunable and switchable magnetic dipole patterns in nanostructured superconductors

Jun-Yi Ge[1,2], Vladimir N. Gladilin[2,3], Jacques Tempere [3], Jozef T. Devreese[3] & Victor V. Moshchalkov[2]

Design and manipulation of magnetic moment arrays have been at the focus of studying the interesting cooperative physical phenomena in various magnetic systems. However, long-range ordered magnetic moments are rather difficult to achieve due to the excited states arising from the relatively weak exchange interactions between the localized moments. Here, using a nanostructured superconductor, we investigate a perfectly ordered magnetic dipole pattern with the magnetic poles having the same distribution as the magnetic charges in an artificial spin ice. The magnetic states can simply be switched on/off by applying a current flowing through nanopatterned area. Moreover, by coupling magnetic dipoles with the pinned vortex lattice, we are able to erase the positive/negative poles, resulting in a magnetic dipole pattern of only one polarity, analogous to the recently predicted vortex ice. These switchable and tunable magnetic dipole patterns open pathways for the study of exotic ordering phenomena in magnetic systems.

[1] Materials Genome Institute, Shanghai University, Shangda Road 99, 200444 Shanghai, China. [2] INPAC-Institute for Nanoscale Physics and Chemistry, KU Leuven, Celestijnenlaan 200D, B-3001 Leuven, Belgium. [3] TQC-Theory of Quantum and Complex Systems, Universiteit Antwerpen, Universiteitsplein 1, B-2610 Antwerpen, Belgium. Correspondence and requests for materials should be addressed to J.-Y.G. (email: Junyi_Ge@t.shu.edu.cn) or to V.V.M. (email: Victor.Moshchalkov@kuleuven.be)

Artificial magnetic moment arrays have been successfully used to study intriguing many-body effects related to spin interactions, such as the spin-glass phase transition[1,2], exchange bias effect[3], and frustration in spin ice systems[4–8]. Among investigated so far magnetic systems, one of the most intensively studied is the artificial spin ice, which exhibits exotic physical properties including the residual entropy at low-temperature limit[6,9], large amount of degenerate states[10], emergent excitations of magnetic monopoles and Dirac strings[11–15].

Moreover, artificial spin ice (ASI) systems provide an ideal platform to study these exotic phenomena under conditions of the controlled design of geometric frustration and real-space observation of local spin configurations[14,16–20]. In a square spin ice four ferromagnetic islands meet at the vertex of a square lattice, and four types of vertex configurations (as well as the corresponding types of magnetic charge distribution) can be formed (Fig. 1b and Supplementary Fig. 1). The spin ice vertex has a twofold degenerate ground state configuration (shown as type-I in Supplementary Fig. 1), which follows the ice rule with two spins pointing inward and two pointing outward at the vertex of a square lattice. Excited vertex configurations in ASI can comply with the ice rule (in which case we refer to them as type-II configurations), or break it: in type-III configurations three spins are pointing in or out, and in type-IV excitations, all spins are pointing in or out of the lattice vertex. The difficulty in studying ASI is that due to the large magnetic energy scales ($10^4$ K) of ferromagnetic islands, it is impossible to use thermal energy to randomize magnetic moments. Experimentally, it has been found that various long-range ordered states are rather difficult to realize through applied demagnetization and thermalization

protocols[14,17,21–25]. A lot of efforts have been devoted to manipulation of the magnetic states of ASI. So far, in square ASI, a sizable ordering of vertices has only been observed in type-I and type-II states, while long-range ordering of type-III and type-IV states has never been observed experimentally due to relatively high interaction energy between the ferromagnetic islands.

To realize long-range ordered ice states, Wang et al.[26] have redesigned an artificial pattern with ferromagnetic islands by focusing on the distribution of magnetic charges instead of spins. This enables a better controllability over magnetic charge distribution and various types of long-range ordering of magnetic charge ice states have been observed. However, the arrangements of spins (ferromagnetic islands) are completely different from those in traditional ASI and no spin ice rule is applicable in the former case. Similar to traditional ASI, the number and the strength of magnetic charges are fixed once the ferromagnetic islands are created, no matter what kind of the order they have. This limits the application of long-range ordered states, for example, as templates to couple with other systems.

Here, to mimic the magnetic charge ice states, we propose a protocol to design geometrical magnetic moment arrays by using magnetic dipoles generated in a nanostructured superconductor. The magnetic dipoles that emulate the macrospins, are created by applying a current around the elongated antidots. This approach allows switching on/off the magnetic dipoles by simply controlling the current. Also, the strength of the magnetic dipoles can be tuned by changing the current density. This tunability was not previously achieved in any kind of spin ice. In our nanostructured superconductor, the generated magnetic states can also be combined with a pinned vortex lattice, thus enabling the

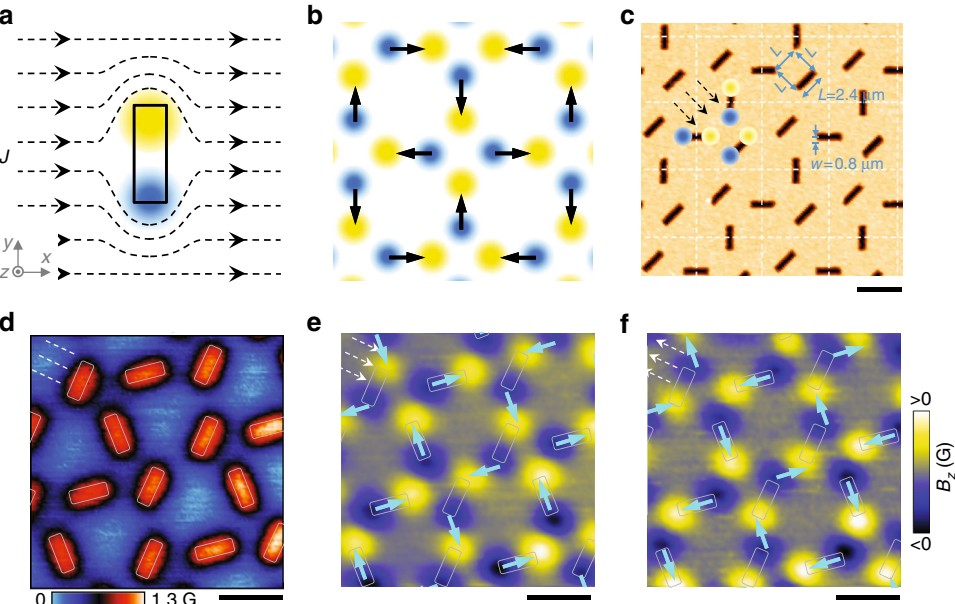

**Fig. 1** Design and formation of the magnetic-charge-ice-like magnetic states. **a** Schematic view of a magnetic dipole generated by applying a current (indicated by dashed lines) around an elongated antidot. The yellow (blue) color indicates positive (negative) magnetic field. **b** Schematics of magnetic charge ice ground state ordering. The arrows, pointing from negative (blue) to positive (yellow) charges and mimicking spins, form the ground state of artificial square spin ice. **c** Atomic force microscopy image showing the topography of the sample. When applying a current in diagonal direction, the ground state magnetic-charge-ice-like pattern distribution can be observed in each square unit ($5.8 \times 5.8\ \mu m^2$) separated by the dashed lines. **d** SHPM image of vortex pattern observed after field cooling to 4.2 K at first matching field 1.23 Oe. White rectangles show the positions of antidots in the scanned area. The dashed lines, parallel to the nearest sample edge, indicate the direction along which Meissner current flows. **e**, **f** Scanning Hall probe microscopy images of magnetic dipole distributions corresponding to the twofold degenerate ground state of a magnetic charge ice. The images were observed after zero-field cooling to 4.2 K and then applying magnetic field of 4.5 Oe (**e**) or −4.5 Oe (**f**). The dashed arrows indicate the directions of the induced Meissner currents. The rectangles mark the positions of antidots. The solid arrows show the ground state of a square spin ice. All scale bars, 4 μm

possibility to erase positive/negative magnetic poles by partial annihilation of the magnetic flux. In such a way, an ordered magnetic pattern analogous to the predicted vortex ice is created. We also demonstrate the possibility of studying vortex ice with our sample design.

## Results

**Design and formation of ordered magnetic moment arrays.** In ASI systems, each macrospin (ferromagnetic island) can be replaced with a dumbbell of magnetic charges, one positive and one negative. Instead of using ferromagnetic islands, in our sample, the magnetic patterns are generated by applying a current that flows around antidots. As schematically shown in Fig. 1a, positive (yellow) and negative (blue) magnetic poles are formed at two ends of an elongated antidot[27,28]. In pyrochlores[6–8] and ASI systems[16,19], the number of spins is fixed no matter what kind of vertex types they form. Compared with other magnetic systems, in our case, a major advantage of using nanostructured superconductor is that the magnetic poles can simply be switched on/off by applying a current. In this paper, instead of using transport current, a Meissner current (Supplementary Note 1), induced by applying an external field (smaller than the lower critical field $H_{c1}$), is used to generate the magnetic poles.

According to the ice rule, the ground state requires the presence of two positive and two negative charges at each vertex. Figure 1b sketches the magnetic charge distribution in a square spin ice (spins are represented by the arrows). In order to form such kind of magnetic state, we follow the protocol by focusing on the magnetic charges instead of spins[26] and design an antidot lattice as shown in Fig. 1c. The elongated antidots were introduced to a superconducting Pb film by using standard e-beam lithography technique (see methods), with the nearest neighbor distance (between magnetic poles) being equal to the antidot length. The chosen antidot size provides a sufficient pinning strength for single-quantum vortices, while at the same time it allows to avoid the formation of multiquantum vortices at low magnetic fields. In Fig. 1c, the antidot lattice is schematically subdivided into several square units shown by the dashed lines. In each square, the ground-state-like magnetic pole distribution is generated by applying a current along the dashed arrows.

Figure 1d represents the vortex lattice observed at first matching field 1.23 Oe, where each antidot is occupied by one vortex. The scanned area is chosen close to the sample edge which is parallel to the dashed lines (Supplementary Fig. 2). When applying a Meissner current through the scanned area after zero-field-cooling, two types of magnetic states, similar to the ground states of magnetic charge ice, are observed, as shown in Fig. 1e, f. Since the polarity of magnetic poles at each antidot only depends on the current flow direction, in principle, a long-range ordered magnetic state can be realized in the whole sample. By introducing different antidot lattices, we are able to produce also the other three types of vertices predicted for square spin ice[14] (Supplementary Fig. 3–5).

**Manipulation of magnetic dipole strength.** In ASI with ferromagnetic islands, it is rather difficult to change the interaction between the macrospins once the lattice parameter is fixed. In Fig. 2, we have shown that, contrary to that, in nanostructured superconductors the intensity of magnetic poles (i.e., the interaction strength) can be tuned by varying the Meissner current density (magnetic fields),which is directly proportional to the external magnetic field at field values smaller than the penetration field[27]. As a result, we can switch the magnetic-charge-ice-like state on (Fig. 2a, b) and then tune the intensity of magnetic poles (Fig. 2b–f). Figure 2g shows the distribution of magnetic field corresponding to one dipole in a square spin ice. The intensity at the center of the positive/negative magnetic poles increases linearly with the magnetic field (Fig. 2h). We would like to mention that such a classical behavior of magnetic poles does not violate at all the fluxoid-quantization rule in superconducting condensate. The magnetic dipoles arise from the redistribution of supercurrent, which does not change the topology of the superconducting condensate. These non-integer magnetic poles and antipoles always appear in pairs as bound dipoles, so that the total flux/fluxoid corresponding to a dipole is zero and therefore remains quantized.

As compared to the known magnetic states of spin ice/magnetic charge ice systems, our design provides a simple and effective way to generate different magnetic states with similar field distributions. More importantly, this design makes

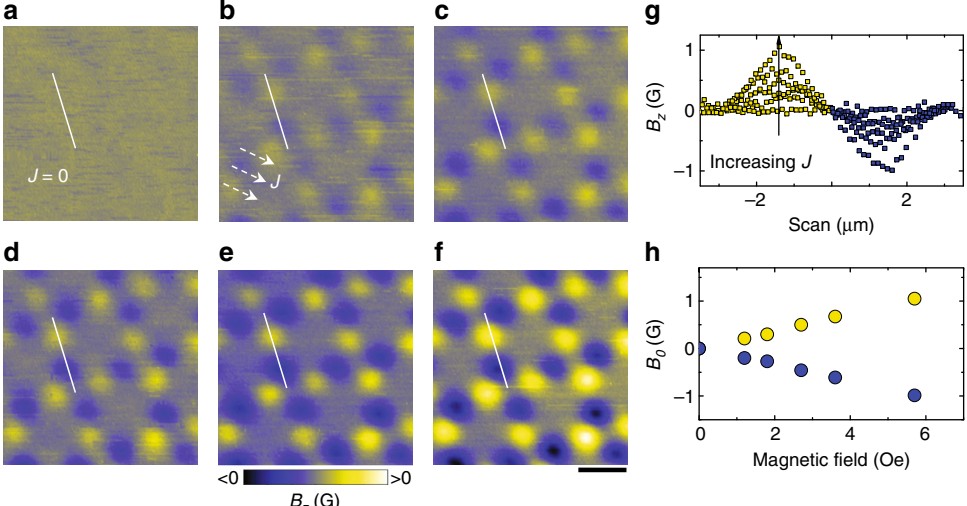

**Fig. 2** Tunable strength of magnetic poles for the magnetic-charge-ice-like patterns. **a–f** SHPM images observed after first performing zero-field cooling to 4.2 K (**a**) and then increasing magnetic field to 1.2 Oe (**b**), 1.8 Oe (**c**), 2.7 Oe (**d**), 3.6 Oe (**e**), 5.7 Oe (**f**). The arrows indicate the flowing direction of the Meissner current. Scale bar, 4 μm. **g** Magnetic field profiles along the solid line shown in **a–f**. **h** The magnetic field $B_0$ at the center of the positive and negative poles, as derived from **g**, exhibits a linear dependence with external magnetic field (current density)

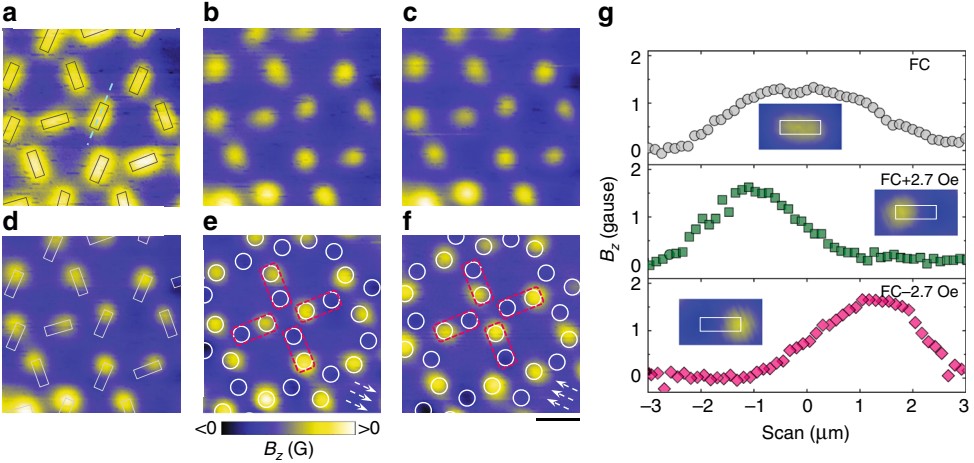

**Fig. 3** Generation of the vortex-ice-like patterns by erasing positive/negative magnetic poles. **a** SHPM images taken after field cooling to 4.2 K at first matching field $H_1$ (**a**), followed by a change of the external field to $H_1 + 0.6$ Oe (**b**), $H_1 + 1.2$ Oe (**c**), $H_1 + 2.1$ Oe (**d**), $H_1 + 2.7$ Oe (**e**), $H_1$-2.7 Oe (**f**). At each antidot, the induced screening current generates a magnetic dipole which overlaps with the magnetic field of a pinned vortex. The negative pole is annihilated while the positive pole is strengthened. In **a** and **d**, the rectangles indicate the positions of antidots. In **e** and **f** the white circles mark the positions where the positive and negative poles are expected to form. Due to the annihilation of negative poles with part of the positive field, corresponding to the pinned vortices, only positive poles are observed. At the vertex formed by four pairs of poles (indicated by dashed rectangles), two magnetic poles sit close to the vertex while the other two are located away from the vertex. The configuration is analogous to the vortex ice state. The arrows indicate the local direction of the supercurrent. Scale bar, 4 μm. **g** Magnetic field $H_1$ along the long axis of an antidot as indicated by the dashed line in **a** at magnetic fields equal to $H_1$ (circles), $H_1 + 2.7$ Oe (squares) and $H_1$-2.7 Oe (diamonds). The insets show the SHPM images of one antidot at the corresponding magnetic fields

various magnetic states switchable and tunable, while in different realizations of spin ice and magnetic charge ice, the number of magnetic charges as well as their intensities are fixed, no matter what kind of order they are in. This adds more flexibility to our magnetic system when using it as a template to create a well controlled potential-energy landscape for other systems, such as ultracold atoms[29,30], superconducting vortices[31,32], graphene[33]. Moreover, our switchable magnetic states only depend on the direction and density of the applied current. For this reason, they are protected against distortions, caused, for instance, by the stray magnetic field generated by vortices.

**Selective magnetic pole erasure by coupling to the vortex lattice.** Another intrinsic feature of the previously studied spin ice and magnetic charge ice systems is that the positive and negative magnetic charges always appear in pairs. The sample remains in a charge neutral configuration. It would be interesting if one could selectively remove the positive/negative magnetic charges, leaving the ordered charge ice state with only one polarity. Recently, such an ice state has been predicted for nanostructured superconductors, where four elongated double-well pinning sites have been arranged to meet at one vertex point of a square lattice[34]. At half matching field, vortices locate at the pinning sites following the ice rule: two vortices sit close to the vertex and the other two vortices sit far away from the vertex. This kind of ice state exhibits interesting properties, such as the anomalous matching effect[35,36], where the critical current density at half matching field is even larger than at zero field. In the following, we will show that such a magnetic state can be realized in our nanostructured superconductors by combining a vortex lattice with magnetic dipoles.

Figure 3a presents the vortex lattice observed after field-cooling at first matching field. Each antidot is occupied by a vortex. By increasing the external field, a screening current is induced inside the superconductor. As a result, a magnetic dipole is formed with positive and negative magnetic charges at the two ends of each

antidot. The positive/negative magnetic poles overlap with the magnetic fields of trapped vortices, weakening the negative magnetic pole at one side of an antidot and, at the same time, making the positive magnetic pole stronger (Fig. 3b, d). With further increasing the supercurrent density, eventually, the negative pole is erased, resulting in a magnetic pole distribution with only one polarity (Fig. 3e). The positions (circles), where magnetic poles are formed, can be paired as indicated by the dashed rectangles. At each vertex point met by three antidots, the magnetic pole distribution follows the vortex ice rule: two close/two far away (Supplementary Fig. 6). When reversing the supercurrent direction, the other ground state of vortex-ice like state can be observed as shown in Fig. 3f. Figure 3g displays the magnetic field profiles along an antidot before (top panel) and after erasing (middle and lower panels) one magnetic pole. A similar vortex-ice-like state with only negative magnetic poles can also be formed by combining an antivortex lattice with magnetic dipole pattern. Moreover, by overlapping the magnetic dipole pattern with a vortex lattice below the first matching field, we are able to selectively erase magnetic poles at the antidots occupied by vortices (Supplementary Fig. 7). Such an approach enables the study of novel vertex defects in a variety of magnetic-charge-ice-like states.

**Frustrated vortex lattice in nanostructured superconductors.** In not-patterned type-II superconductors, vortices usually have circular cores. The interaction between two vortices is purely repulsive. Under such repulsive interaction, vortices form triangular lattice. However, for vortices with elongated cores, the interactions become anisotropic. Frustration in the vortex-vortex interaction arises from the elongation of vortex cores in the direction of the closest neighbor[37–39]. This leads to the appearance of an attractive interaction between vortices. As a result, vortices tend to form stripes instead of triangular lattice. In our current work, the trapped vortices were forced to follow the

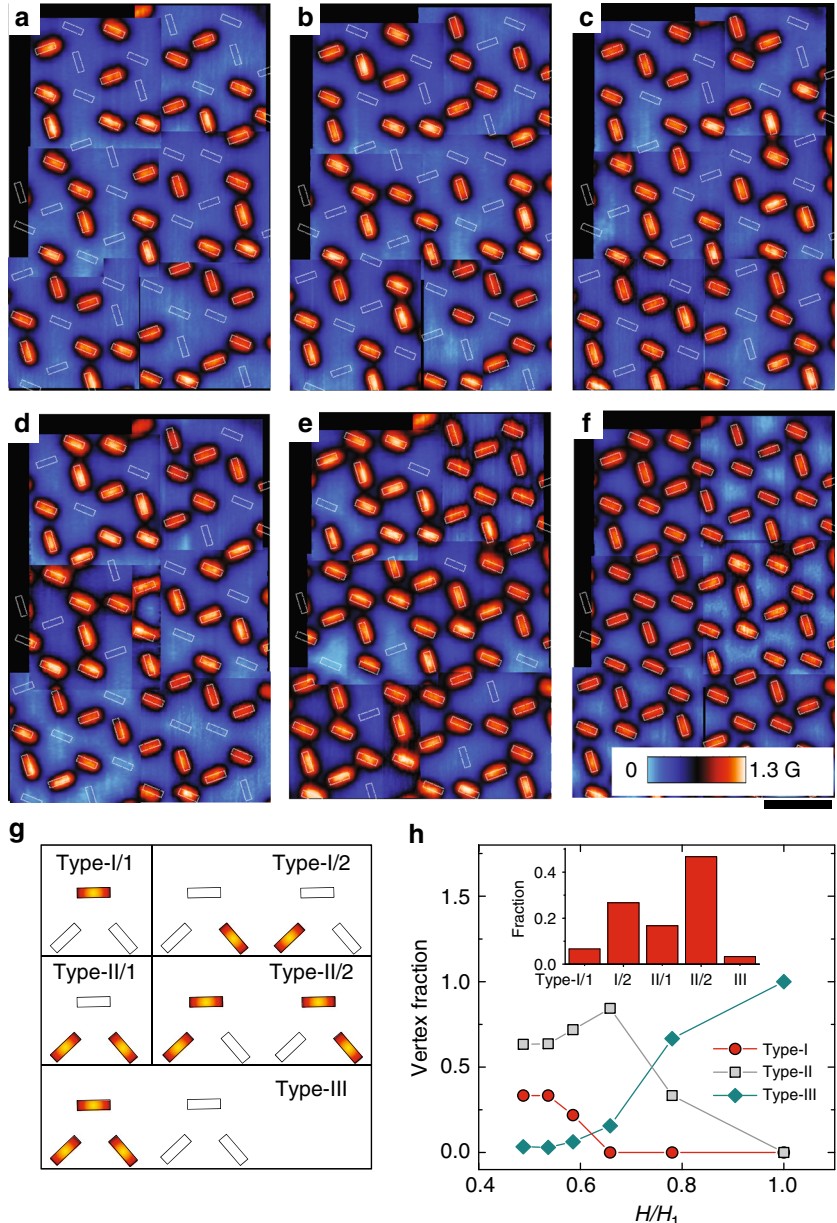

**Fig. 4** Vortex lattice evolution with magnetic field. **a–f** Vortex lattice observed at 4.2 K and different applied magnetic fields $0.49H_1$ (**a**), $0.54H_1$ (**b**), $0.59H_1$ (**c**), $0.66H_1$ (**d**), $0.78H_1$ (**e**) and $H_1$ (**f**). Scale bar, 4 μm. **g** Schematic view of various unit configurations. **h** Statistics of vertex configurations in vortex ice. The inset histogram shows the distribution of various types at $H = 0.49H_1$

elongated antidot geometry. As a result, frustrated interactions occur together with the geometric frustration. The two effects combined lead to the formation of intricate vortex patterns. Therefore, exotic vortex patterns, such as stripes, are expected to form. In the last part, we briefly study the possible vortex ice states in our designed nanostructured superconductor.

Each vortex ice unit (squares in Fig. 1c) contains three antidots. In total, eight vortex configurations can form, as shown in Fig. 4g. We have seen that close to half matching field (Fig. 4a), most unit cells follow the one-occupied/two-empty or two-occupied/one-empty principle, which is similar to the quasi-ice rule in a kagome spin ice lattice[40]. Under such a rule, vortices tend to form stripes (Fig. 4a–d), which are unlikely to intersect with each other. Above $2H_1/3$ (Fig. 4e), the fraction of units with type-III vortex configuration increases and eventually the quasi-ice rule is destroyed (Fig. 4f). Figure 4h summarizes the fraction of

various vertex configurations as a function of magnetic field. Our results suggest that the designed antidot lattice might also be used for further advanced studies of vortex ice in nanostructured superconductors.

In summary, we have fabricated a nanostructured super-conductor in which the long-range ordered ground states of square magnetic charge ice can be mimicked by applying a supercurrent. Following a similar protocol, various long-range ordered magnetic states can be implemented. By changing the current density, one can fine tune the intensity of magnetic poles, which is not possible for the magnetic charges in pyrochlores and nanomagnet spin ice systems. Moreover, by combining the magnetic dipole pattern with a vortex lattice, the positive/negative magnetic poles can be selectively erased from the system, resulting in a magnetic dipole distribution similar to a vortex ice pattern. Such a possibility opens new ways

to deliberately introduce defects to the ordered artificial magnetic systems, which is particularly interesting when using the systems as templates to study defect formation as well as dynamics and relaxation of the coupled systems. We also demonstrate the ability of studying the frustration in the interaction of vortices with elongated cores in our nanostructured superconductor.

## Methods

**Sample**. The antidot lattice pattern was fabricated by electron beam lithography using a Raith e-beam system. A superconducting Pb film of 90 nm thickness was then electron-beam evaporated in ultrahigh vacuum ($3 \times 10^{-8}$ Torr) with a rate of $1 \, \text{Ås}^{-1}$. The substrate was cooled to 77 K by liquid nitrogen to ensure a homogeneous growth of the Pb film. A 10 nm Ge capping layer was deposited on top of the Pb film to protect it from oxidation. The liftoff was performed on a double layer of polymethyl methacrylate (300 nm). After liftoff, the sample with a dimension of $200 \times 200 \, \mu\text{m}^2$ was transferred to a sputtering machine, where it was covered with a 35 nm layer of Au which plays the role of conducting layer for the tunneling junction of the scanning tunneling microscope (STM) tip of the Hall probe. $T_c = 7.35$ K was determined by directly imaging the vortex with increasing temperature (Supplementary Fig. 8). The penetration depth and coherence length were estimated to be 151 nm and 52 nm, respectively (Supplementary Note 2). The Ginzburg-Landau parameter is $\kappa = 2.9$, indicating that our sample is a type-II superconductor.

**Scanning Hall probe microscopy**. The sample was imaged by using a low-temperature scanning Hall probe microscope in the lift-off mode. The Hall probe was first brought to close proximity of the sample surface by using an STM tip, which is asssembled together with the Hall cross. Then the probe was lifted 300 nm. This ensures a non-invasive measurement of the magnetic field pattern. In all the measurements, the applied magnetic field is perpendicular to the surface of the sample. The induced in-plane Meissner current density is $J(x) = -2Hx/\sqrt{w^2 - x^2}$, where $w$ is the half width of the superconducting stripe, and $x$ is the distance from its axis (Supplementary Note 1).

**Data availability**. The data that support the findings of this study are available from the corresponding authors upon request.

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

## Acknowledgements

We thank J. Van de Vondel and V. S. Zharinov for the help of making the nanostructured pattern. J.-Y.G., V.N.G and V.V.M. thank the support from the Methusalem funding by the Flemish government, the Flemish Science Foundation (FWO) and the COST action COLHYBYI. J.-Y.G. also thanks the support by The Program for Professor of Special Appointment (Eastern Scholar) at Shanghai Institutions of Higher Learning. J. T. acknowledges the support from the Research Council of Antwerp University (BOF), and from the Flemish Science Foundation (FWO) Grant No. G.0429.15 N.

## Author contributions

J.Y.G. made the nanostructured sample and performed the SHPM measurements. J.Y.G. wrote the manuscript. All authors contributed to the discussion and analysis of the results. V.V.M. coordinated the whole work.

## Additional information

**Competing interests:** The authors declare no competing interests.

