## [Peer Review File · Nature Communications]

Reviewers' comments:

Reviewer #1 (Remarks to the Author):

This paper presents an experimental study of an interesting nanostructured superconducting system. Using the fact that an elongated antidot will trap a magnetic dipole when applying a current, the authors proposed and fabricated antidot lattices that realize regular patterns of magnetic charges. They further demonstrate the tenability of magnetic charge strengths and polarities.

Despite the fact that the authors presented their results using the terminology of artificial spin ice or ice systems in general (they also made frequent comparison with artificial ice systems), I don't think this work has anything to do with spin-ice or frustrated magnets (except maybe the last part of the paper, related to Figure 4).

In such setup, the polarity of the magnetic dipole is fixed by the super-current, as demonstrated in Fig. 1a. So the magnetic dipole here is not even a dynamical variable, not to mention their long-range ordering. That is why I used the term "regular patterns of magnetic charges" above, instead of "long-range ordering of magnetic charges".

In my opinion, this is a very interesting work to realize arrays of magnetic charges. Such array might be useful for other applications with further engineering. And the fact that the basic units here have magnetic charge degree of freedom is a novelty of the proposed system. However, this is not spin-ice physics.

There is some many-body physics at the end of the paper. But, due to the very limited results there, more detailed characterization of the frustrated interactions is required.

Based on the above points, I do not recommend the publication of this paper in Nature Communications. The authors might consider more engineering-oriented journals. And in any case, I strongly suggest the authors remove the comparisons with artificial ice systems. It seems to me these comparisons are superficial and misleading.

Reviewer #2 (Remarks to the Author):

This manuscript describes elegantly artificial long-range ordered magnetic charge ice lattices by controlling the Meissner current and vortices in nanoengineered antidot lattices of superconducting films in a very intelligent way. This strategy allows them to define a ground state, switch on/off selected magnetic charges and therefore design distributions of ice patterns. The paper opens the field to additional ordered arrays that can be engineered by modifying the antidote lattice and therefore overall widens the field. The manuscript is addressed to a wider community beyond superconductivity which are interested in the control of frustration systems and exotic orders. The SHPM experiments are unique and with a very good quality and they made a good partnership with experts in simulations. Therefore, I am convinced that this paper should be published in Nature Communications. I have just few comments that should be attained for the completeness of the manuscript, specially addressed to the wider community beyond superconducting mesoscopics or to be a self-sustained article.

1. Mention which is the value of the matching field for the specific configuration used. Also mention the dimension of the square lattice (dashed white lines) to make it easier to the reader.
2. Addressed beyond the superconducting audience, comment on the importance of dimensions of antidots in comparison with critical parameters of the superconducting material chosen. Comment the relevance of being in a mesoscopic range.

3. Indicate dimensions of the antidots to reach the different configurations shown in the supplementary information.
4. Comment the chosen conditions used like that of the nearest neighbour distance being equal to the antidot length.
5. They should mention the simulations framework used and the theoretical hypothesis done to obtain the resulting simulated arrays. A methods section addressed to the simulation scheme should be incorporated.
6. They should indicate the range of current values that are used to generate the charge state and if they are the same used to study the system above the first matching field
7. They should indicate the samples dimensions and how they inject the current.

Reviewer #3 (Remarks to the Author):

Referee Report

J-Y Ge et al.

NCOMMS-17-30214

This paper reports the fabrication and characterization (via scanning Hall probe—SHP) of the magnetic structure of a novel type of “tunable magnetic charge ice”, which basically consists of a thin, superconducting Pb film (passivated by a Au/Ti bilayer) patterned with an antidot lattice composed of elongated rectangular holes. Various pattern geometries are considered, which modify the spatial distribution of magnetic flux, and the ways that flux can be manipulated by small externally applied magnetic fields.

The abstract states that the authors “investigate a perfectly ordered ground state of artificial magnetic charge ice realized in a superconductor with nanoengineered antidot lattice. The magnetic charge ice can then simply be switched on/off by applying a current flowing through nanopatterned area, where dipoles with tunable magnetic charges are generated. Moreover, by coupling magnetic dipoles with the pinned vortex lattice, we are able to erase the positive/negative charges, which results in a magnetic charge lattice pattern of only one polarity, which is analogous to the recently predicted vortex ice. This switchable and tunable magnetic charge ice opens new ways for the study of exotic ordering phenomena in ice systems.”

I believe that this paper does report a novel type of “magnetic dipole ice” (my term) that behaves in interesting new ways, especially by applying small magnetic fields that generate Meissner currents near the edges of the superconducting, patterned Pb film. The possible experimental realization (see SM Figs. s3 and s4) of pure Type III and Type IV square ASI charge states, as well as demonstrated states with net “magnetic charge”, are particularly intriguing and innovative.

The submitted manuscript does a (deceptively) good job of simplifying a very complicated system for a general audience, but have oversimplified their results in the case of expert readers. As I will argue below, there are serious omissions in the submitted manuscript which bear upon the robustness of the observed magnetic textures and the physics that underlie the SHP images obtained by the authors. Hopefully, many of these technical issues can be adequately covered by additions to the Supplementary Material (SM), while preserving a short paper worthy of publication in *Nature Communications*.

I. Problems with Spin Ice Terminology

A. Artificial spin ice (ASI) has been a subject of interest since the early work of Schiffer’s Group in 2006, and a corresponding set of technical terms has emerged over the years. However, these terms

are currently evolving in response to a recent increase in the number and types of artificial (mesoscale) materials fabricated to exhibit frustration (see the *Physics Today* article by Ian Gilbert et al. in 2016). This has led to a generalization of ASI terms that is not yet settled. The present authors should therefore be very careful in using old terms, especially since they show their new “ice” involves underlying interactions or physics that differ from older ASI examples in the literature.

In the context of frustrated metamaterials, the term, “ice” originally referred to frustration indicated by two possible lengths of oxygen-hydrogen bonds between tetrahedrally coordinated oxygen atoms, and the large entropy observed for cubic water ice at low temperatures. An analogous case of frustrated magnetism (including large low temperature entropy) was subsequently observed in tetrahedrally coordinated rare earth spins in pyrochlore structures by Ramirez and coworkers; here, degenerate configurations where two of four rare earth spins pointed into a tetrahedron, and two pointed outward, led to their designation as “spin ices”. This idea was further extended to “artificial spin ices” (ASI) by Schiffer and coworkers in 2006: They fabricated square ASI composed of square arrays of elongated ferromagnetic dots of mesoscopic dimensions whose shape anisotropy caused them to behave as Ising dipoles. Most relevant, these classical Ising spins could be directly imaged by MFM and PEEM, which showed they obeyed a “two-in/two-out, Pauling ice rule” that governs the lowest energy spin arrangements of the square ASI (see the SM Fig. s1).

The term “magnetic charge ice” conventionally refers to regions of space (occupied by ferromagnetic nanostructures) in which the ***divergence of the magnetization is non-zero***, as in the case of either of the two ends of elongated segments of uniformly magnetized ferromagnetic films (e.g., permalloy). Consider a Gaussian “pill box” with one end inside the ferromagnetic dot and the opposite end outside the ferromagnetic dot, which yields a nonzero magnetic charge density $\rho = -\text{div}\mathbf{M}$ at both end surfaces of the dot. The corresponding magnetic charge is found from Gauss’ theorem: $Q = \int \mathbf{M} \cdot \mathbf{n} \, dA = MA$ (in terms of the surface normal \mathbf{n} , the uniform magnetization \mathbf{M} , and the end area A of the dot). Note that this magnetic charge is defined as the source charge for the magnetic field intensity \mathbf{H} , not the magnetic induction \mathbf{B} (which retains its fundamental zero divergence). The idea of magnetic charge is very important in ASI, since it leads to highly simplified Coulomb (“dumbbell”) models for quantitative estimates of magnetic energies and degeneracies (entropies) of various arrangements of Ising moments.

In the present case, the authors state at the bottom of page 1 of the manuscript that they “design a new artificial ice state by using a nanostructured superconductor which introduces better controllability of ***magnetic charge order***” (my italics). The nature of this “charge order” is introduced in Figure 1. In particular, Fig. 1a is intended to show how “magnetic charge” is defined in this paper.

An Aside: It would help the reader if the signs of the fields and “charges” were defined in terms of a z -axis. I assume that the figures depict the x - y plane, and the undefined z -axis points into the paper.

Figure 1a tries to define a “positive” magnetic charge as indicated by a yellow blob, and a “negative” magnetic charge indicated by a blue blob, in order to correspond directly with the SHP data presented in the paper. The reader gradually becomes aware that the colored blobs refer to the ***magnetic field intensity*** generated by the supercurrent density \mathbf{J} that is compelled to flow around the antidot, and that Ampere’s law is applied to define the yellow or blue blobs that represent “magnetic charge”. This is where the paper starts to lose clarity, because the ***magnetic induction*** generated within the antidot volume by the supercurrent obeys $\text{div}\mathbf{B} = 0$. Moreover, the magnetization $4\pi\mathbf{M} = \mathbf{B} - \mathbf{H} = 0$ inside the empty antidot, so we must have $\text{div}\mathbf{H} = 0$ inside the antidot, which demands that ***the magnetic charge is zero throughout the antidot***. On the other hand, the SHP measures a non-zero magnetic induction \mathbf{B} directed along the z -direction, as shown in

Figs. 1e or 1f, which agrees with Ampere's law. I conclude that **the yellow and blue blobs cannot represent "magnetic charges" as they are defined in the current ASI literature.**

B. If one asks, "where is the divergent magnetization?", one is reminded that that picture is best suited for a magnetostatics problem, not for a case of a variable source current \mathbf{J} . This leads to another set of questions concerning the detailed nature of $\mathbf{J}(\mathbf{r})$, which is not well-enough defined/discussed in the paper, given that it is a source of the blue and yellow blobs seen in SHP. There is also an inadequate discussion of the involvement of magnetic flux pinning by the antidot lattice, which is only vaguely introduced Fig. 1d. The blobs shown in various figures evidently represent **z -components of field** generated by **two source currents**, one due to Meissner "edge currents" (see Fig. 1a), and one due to trapped/pinned "vortices" (authors' terms) in the antidots via field cooling (see Fig. 1d). These two contributions need to be better explained in the early part of the paper and/or in the Supplementary Material (SM).

What I view is an incorrect use of the term, "magnetic charge" is compounded many times over in the discussion at the bottom of page 2 of the manuscript. There is also no discussion or clarification of the fact that the **total magnetic flux must be quantized in the region of each antidot**, according to the arguments originally given by London (see *Introduction to Superconductivity*, by M. Tinkham, page 122). The London picture demands that a close coupling (constraint) exists between the local supercurrent density $\mathbf{J}(\mathbf{r})$ and the number of **quantized fluxoids** trapped by a given antidot. The proximity of the supercurrent flow to any antidot shown in Fig. 1 complicates derivation of an analytic expression for either $\mathbf{J}(\mathbf{r})$ (the source of the superconducting **magnetization** of interest, and therefore the "magnetic charge" present in the same region of space) or the spatial extent of the magnetic field intensity inside/near a given antidot.

C. On page 1, third paragraph, the authors refer to the work of Kwok's Argonne Group on "rewriteable artificial spin ice" which focuses on magnetic charge order rather than Ising spin order in a modified ASI [27]. The authors state that "no spin ice rule is applicable here". This statement should be briefly clarified, since this work is seminal to the present paper.

D. On page 1, the fourth paragraph states, "The magnetic dipoles that emulate the spins, are generated by applying a current around elongated antidots. Such a way allows switching on/off the magnetic dipoles by simply controlling the current. Also, the strength of the magnetic dipoles can be tuned by changing the current density..." I assume "magnetic dipoles" is a term referring to either a single, or two oppositely directed dipoles in Fig. 1. I suggest the authors realize the wisdom of their own wording here, and use some variation of "dipole" (instead of "magnetic charges") to describe the CW and CCW source currents (Fig. 1a) that generate the yellow and blue blobs, respectively, that represent the magnetic field concentrations along the z -direction in Figs. 1 and 2. The true source currents $\mathbf{j}(\mathbf{r})$ that generate the "dipoles" must flow within the x - y plane inside the superconductor, and are not located at the positions of the yellow and blue blobs **inside** an empty antidot.

E. The authors remark in the second paragraph on page 2, "Instead of using ferromagnetic islands, in our sample, the magnetic charges are generated by applying a current that flows around antidots. As schematically shown in Fig. 1a, positive (yellow) and negative (blue) magnetic charges are formed at two ends of the elongated antidot [28, 29]." Given my arguments in **D** above, this usage of "magnetic charge" is misleading (and wrong).

One might try to emulate the field distribution of the yellow and blue blobs by imagining two rectangular current loops (dipoles) placed at opposite ends of the rectangular antidot such that their currents and magnetic fields cancel near the midpoint of the antidot. That is my crude suggestion for an analogue of the "dumbbell charge" model for ASI.

F. On page 2 of the manuscript, a “pinned vortex lattice” refers to Fig. 1d, which shows an SHP image of magnetic flux trapped by field cooling through T_c . I recommend the authors (optionally) adhere to the term “fluxoid” here and elsewhere to avoid confusion with the quantized “Abrikosov vortex” present in the Type II mixed state, or “multiquantum vortices” found in antidot lattices in Type I materials (see G.R. Berdiyrov, M.V. Milosevic, F.M. Peeters, *Physica C* 437–438 (2006) 25). The authors also refer to “recently predicted vortex ice” in the abstract, and “vortex ice” on page 2 of the manuscript, which is not adequately defined or referenced (it is only vaguely mentioned in the lead paragraph on page 1, with referencing [7-9]).

II. Problems with Sample Characteristics

There are several ambiguities regarding the nature and behavior of the patterned Pb films used in this study. Some are probably trivial, requiring only brief clarification in either the main text or the SM; others bear on the robustness of the ice behavior and how it is controlled.

A. The samples consist of Pb(90nm)/Ge(10nm)/Au(35nm) heterostructures with unstated lateral dimensions (x-y plane). The superconducting transition temperature $T_c = 7.35$ K, which differs slightly from the accepted bulk Pb $T_c = 7.190$ K. The web values of the relevant bulk Pb parameters are coherence length $\xi_0 = 83$ nm, London penetration depth $\lambda_L = 37$ nm. We therefore expect an effective, thin-film penetration depth $\lambda_{\text{eff}} = \lambda_L(\xi_0/t)^{1/2}$, where t is the film thickness in cases where $t \ll \xi_0$, which indicates near-bulk behavior is expected for these samples, since $t = 90$ nm $> \xi_0$. These numbers also indicate that a corresponding unpatterned Pb film is a Type I superconductor, since the bulk Ginzburg-Landau $\kappa \approx \lambda_L/\xi_0 = 0.45 \ll 0.7$. This parameter check should be added to a sample description (so far missing) in the SM.

B. Note that the characteristic lengths given above are zero-temperature values; the effective penetration depth and coherence length increase quite rapidly as temperature approaches T_c from below. The SHP images shown in Fig. 2 were apparently taken at $T = 4.2$ K (Figs. 2g, 2h); information regarding measurement temperatures is insufficient in the paper, and needs to be added in several places, as variations of temperature could drastically alter the observed behaviors.

C. The authors briefly point out that their SHP imaging was performed near the sample edge, due to the spatial distribution of the Meissner screening current: “The scanned area is chosen close to the sample edge which is parallel to the dashed lines” (page 3, first paragraph). It is unfortunate that the authors have not supplied an accurate sketch of the **entire sample film**, showing the geometry of the antidot lattice and its distance from the sample edges; indeed, the magnitude and geometry of the screening Meissner currents generated by applied field variations are the drivers of the magnetic ice configurations shown in Figs. 1 and 2. The dimensions of the antidot lattice include an approximate separation between antidots of 2400 nm (see Fig. 1d), which, with the above parameter check, implies the supercurrent and local fields can exhibit strong spatial dependence (depending on temperature) over the dimensions shown in Fig. 1. The equation given for the spatial dependence of $J(x)$ in the Methods section lacks a reasonable sketch or graph, and says nothing about the effects of the antidot lattice on $\mathbf{J}(\mathbf{r})$, which could be quite complicated. A thorough discussion of such information would be very helpful to the expert reader, and should be added to the SM.

D. The Leuven Group and the Antwerp Theory Group of Peeters have published extensive work on the behavior of Type I films patterned with antidot lattices, and verified the existence of “multiquantum vortices” and other interesting complications due to finite sample and antidot lattice dimensions, temperature, applied field, and so on, which are relevant to the present work.

- 1) What kinds of restrictions (especially regarding temperature and field/current penetration) do these details put on the measurements of magnetic ice behavior reported in this paper?
- 2) How homogeneous is the sample magnetic texture across the *entire sample*?
- 3) As the authors suggest in their introduction, dynamics and poor equilibration have presented serious problems in studies of artificial spin ices; do they complicate the characterization of the equilibrium properties of the present materials? Is the magnetization map of the sample stable *with time* after increments in the applied DC field; that is, are there magnetic relaxation effects and/or hysteresis due to domain wall motion in the (expected) intermediate state, or flux creep in a (unexpected) Type II mixed state?

E. On page 2, second paragraph, the authors state, "In this paper, instead of using transport current, a Meissner current, induced by applying an external field (smaller than the lower critical field H_{c1}), is used to generate the magnetic charges." On page 3, first paragraph, "Figure 1d represents the vortex lattice observed at first matching field, where each antidot is occupied by one vortex. The scanned area is chosen close to the sample edge which is parallel to the dashed lines." On page 3, second paragraph, we read, "the intensity of magnetic charges (i.e. the interaction strength) can be tuned by varying the Meissner current density (magnetic fields), which is directly proportional to the external magnetic field at field values smaller than the penetration field [28]."

This wording implies that the authors see a need to avoid intermediate state domain formation, or other complications involving the sample edges or interior. Why not use transport current to control the ice behavior? These issues should be at least acknowledged and better explained for the expert reader.

F. On page three, third paragraph, it is stated, "the switchable magnetic-charge-ordering state only depends on the direction and density of the applied current. For this reason, it is protected against distortions, caused, for instance, by the stray magnetic field generated by vortices."

Are the authors making a comparison with the Type II mixed state with Abrikosov vortices here? Can possible interstitial "multiquantum vortices" in the Type I intermediate state be relevant here? How does flux quantization affect things? Please clarify this wording.

III. Other Remarks

1) A better discussion of "frustration" in the present context would be useful for the general reader: How do the authors define/quantify frustration in their new ice? This should be associated with a macroscopic (i.e., proportional to system size) residual entropy associated with degeneracies among dipole distributions shown in Fig. 4, for example. Figure 4 and associated text are poorly motivated in the current manuscript (just "tacked on" at the end of the paper). Why were Figs. 4g and 4h shown? This should help clarify the discussion of ice degeneracy for a non-expert.

2) I found the discussion of the proposed relationship of the authors' dipole ice with effects of "elongated cores" and the "anomalous matching effect" in nanostructured superconductors to be vague and therefore not very useful for readers. Additional explanations might add weight to these possibilities.

3) Figure 1d: The white arrowheads are missing, and there is not color scale shown.

4) Figure 4g: The square cell line for Type I/1 is displaced.

IV. Summary

I believe this paper, if carefully re-written, might merit publication in *Nature Communications*. This assumes that my criticisms can be answered in the SM, although there also should be a number of modifications made to the text, in the interest of clarity of exposition and consistent terminology. The main experimental results of the paper demonstrate the authors have fabricated a novel type of frustrated metamaterial that I would prefer to describe as a “superconducting dipole ice”, or some related name. I reluctantly, but strongly object to the authors’ use of the term “magnetic charge ice” and related terms in this paper, as described in my report.

On the other hand, the SHP images reveal an obvious and well documented “dipole ice topology” for the distribution of magnetic induction inside the antidots. The yellow and blue blobs clearly map onto the previously studied magnetic charge distributions for square ASI and rewriteable ASI (for which magnetic charges also provide a quantitative model for the approximate energies and entropies associated with frustrated behavior). The authors might consider the view that the real novelty of their work is that they have documented aspects of “ice behavior” in a patterned superconducting film which cannot be explained in terms of widely applied magnetic charge models for ASI. Moreover, their method of controlling the magnetic textures and dipole site occupancy via external DC field is an interesting advance in the field of frustrated metamaterials.

I am not convinced that ASI “dumbbell” charge models and the like can yield any clear, quantitative understanding of this new system where extended supercurrent sources dominate the behavior. This may require sophisticated numerical simulations based upon Ginsburg-Landau theory. The similar topologies of the induction textures in the present dipole ice system and the magnetic charge maps in square ASI nevertheless suggest that the existing structure and figures for this paper can be re-cast into a *Communication* (with an extended SM file) that clearly and carefully describes novel “ice behavior” without contradicting the existing terminology commonly used in the ASI community and basic magnetostatics text books.

We would like to thank the referees for their suggestions and comments which we found helpful to improve our manuscript. In the following, we address one by one the criticisms raised by all the referees.

Reply to the comments of Reviewer #1

1. This paper presents an experimental study of an interesting nanostructured superconducting system. Using the fact that an elongated antidot will trap a magnetic dipole when applying a current, the authors proposed and fabricated antidot lattices that realize regular patterns of magnetic charges. They further demonstrate the tenability of magnetic charge strengths and polarities.

Despite the fact that the authors presented their results using the terminology of artificial spin ice or ice systems in general (they also made frequent comparison with artificial ice systems), I don't think this work has anything to do with spin-ice or frustrated magnets (except maybe the last part of the paper, related to Figure 4).

In such setup, the polarity of the magnetic dipole is fixed by the super-current, as demonstrated in Fig. 1a. So the magnetic dipole here is not even a dynamical variable, not to mention their long-range ordering. That is why I used the term "regular patterns of magnetic charges" above, instead of "long-range ordering of magnetic charges".

In my opinion, this is a very interesting work to realize arrays of magnetic charges. Such array might be useful for other applications with further engineering. And the fact that the basic units here have magnetic charge degree of freedom is a novelty of the proposed system. However, this is not spin-ice physics.

There is some many-body physics at the end of the paper. But, due to the very limited results there, more detailed characterization of the frustrated interactions is required.

Based on the above points, I do not recommend the publication of this paper in Nature Communications. The authors might consider more engineering-oriented journals. And in any case, I strongly suggest the authors remove the comparisons with artificial ice systems. It seems to me these comparisons are superficial and misleading.

Response:

We agree with the referee that our regular magnetic dipole arrays do not arise from the magnetic interactions between frustrated elemental units (cf. the spin ice systems). In the later case, long-range ordered magnetic states are rather difficult to achieve, which limits the applications of spin ice system, e.g., as substrates to couple with other systems. The main message of our current work is that we develop an alternative way to produce magnetic patterns similar to the magnetic field distributions of long-range ordered spin ice states.

Since the use of the spin-ice terminology, defined in the literature, in application to our system, indeed, can be confusing, we have revised our manuscript as follows.

The title has been changed to “*Tunable and switchable magnetic dipole patterns in nanostructured superconductors*”. We have rewritten the manuscript by removing the terminology of magnetic charge ice when talking about the observed regular magnetic patterns. Also we have replaced “magnetic charges” with “magnetic poles” when talking about the positive/negative magnetic field spots generated by our antidot lattice. We emphasize that, with our method, we are able to generate a magnetic dipole pattern with an “*ice-like*” distribution of magnetic field. The whole manuscript now mainly focus on the tunability and controllability of such magnetic moment arrays.

We believe that such controllable magnetic field distributions are crucial for the development of new research lines in optics, electronics and spintronics as well as for engineering of devices for field modulation. One example is the use of these systems as templates to couple with other magnetic systems, such as artificial spin ice, superconductors or ferromagnetic layers. Another example would be to apply this system to trap ultra cold atoms in quantum simulators. By controlling the geometrical distribution of antidots, various shapes of magnetic traps can be realized.

Reply to the comments of Reviewer #2

This manuscript describes elegantly artificial long-range ordered magnetic charge ice lattices by controlling the Meissner current and vortices in nanoengineered antidot lattices of superconducting films in a very intelligent way. This strategy allows them to define a ground state, switch on/off selected magnetic charges and therefore design distributions of ice patterns. The paper opens the field to additional ordered arrays that can be engineered by modifying the antidote lattice and therefore overall widens the field. The manuscript is addressed to a wider community beyond superconductivity which are interested in the control of frustration systems and exotic orders. The SHPM experiments are unique and with a very good quality and they made a good partnership with experts in simulations. Therefore, I am convinced that this paper should be published in Nature Communications. I have just few comments that should be attained for the completeness of the manuscript, specially addressed to the wider community beyond superconducting mesoscopics or to be a self-sustained article.

1. Mention which is the value of the matching field for the specific configuration used. Also mention the dimension of the square lattice (dashed white lines) to make it easier to the reader.

Response:

The matching field of the nanostructured superconductor is 1.23 Oe. The dimension of the square lattice in Fig. 1c is $5.8 \times 5.8 \mu\text{m}^2$. We have added this information in the revised manuscript.

2. Addressed beyond the superconducting audience, comment on the importance of dimensions of antidots in comparison with critical parameters of the superconducting material chosen. Comment the relevance of being in a mesoscopic range.

Response:

The maximum number (saturation number) of vortices that can be trapped in an antidot (circular geometry) can be estimated as $N_s = a/2\xi$, where a is the size of antidot and ξ is the coherence length. On one hand, if the dimension of the antidot is much smaller than ξ , the pinning force is too weak to effectively trap a vortex at the antidot position. On the other hand, if the size of the antidot is much bigger than the coherence length, then more than one vortices can be trapped by such antidot. Both of the above cases are not suitable for the design and manipulation of "magnetic charge ice-like patterns" as shown in our manuscript. The ideal size of an antidot should be comparable to the coherence length of the superconductor.

As suggested by the referee, we have commented on this point in the revised manuscript.

3. Indicate dimensions of the antidots to reach the different configurations shown in the supplementary information.

Response:

We have now indicated the dimensions of the antidots in the supplementary material. We would like to emphasize that the schematic figures shown in the supplementary material is just an idea to generate different configurations of magnetic charge ice states. The dimensions of the antidots can vary to form charge ice states with different lattice parameters or nearest neighbor distances.

4. Comment the chosen conditions used like that of the nearest neighbor distance being equal to the antidot length.

Response:

In artificial spin ice and artificial vortex ice systems, the most studied geometry is the square lattice where the distance between nearest neighbors is equal to the nanomagnet/paired antidot size [see, e.g., Science 352, 962-966 (2016); PRL 111, 067001 (2013); PRL 102, 237004 (2009)]. The main goal of the present work is to find a way to generate a magnetic charge distribution similar to the artificial spin ice systems. Our specific choice of the antidot lattice geometry is aimed to facilitate comparison with artificial spin ice and artificial vortex ice systems described in the literature.

5. They should mention the simulations framework used and the theoretical hypothesis done to obtain the resulting simulated arrays. A methods section addressed to the simulation scheme should be incorporated.

Response:

We did not perform simulations specifically for the antidot arrays considered in the present manuscript. The distributions of currents and fields shown in Figs. 1(a) and s3 to s5 are schematic drawings, based to a great extent on the distributions obtained earlier in our TDGL simulations [27,28] for antidots and other defects surrounded by Meissner current.

6. They should indicate the range of current values that are used to generate the charge state and if they are the same used to study the system above the first matching field

Response:

The current (Meissner current) used in our research is induced by applying an external magnetic field (smaller than the vortex penetration field). The current density close to the edges is lower than the critical value. As shown in Figs. 1 and 2, all the charge-ice like images are observed in the Meissner state. In Fig. 3, after field-cooling at first matching field, the shielding current that flows along the sample edges is rather weak

since there are a substantial amount of vortices trapped inside the superconductor. When further changing (increasing/decreasing) the external field, a supercurrent will be induced which behaves similar to the Meissner current.

7. They should indicate the samples dimensions and how they inject the current.

Response:

The sample used in the current research has a square geometry with the dimensions of $200 \times 200 \mu\text{m}^2$. The supercurrent (Meissner current) that is used to generate the magnetic dipoles is induced by applying a relatively low external magnetic field. Now we have explicitly indicated this in the manuscript. Also, in the supplementary material, we have added an optical image (Fig. s2) clearly showing the dimensions of the sample and the flowing current direction.

Reply to the comments of Reviewer #3

I believe that this paper does report a novel type of “magnetic dipole ice” (my term) that behaves in interesting new ways, especially by applying small magnetic fields that generate Meissner currents near the edges of the superconducting, patterned Pb film. The possible experimental realization (see SM Figs. s3 and s4) of pure Type III and Type IV square ASI charge states, as well as demonstrated states with net “magnetic charge”, are particularly intriguing and innovative.

The submitted manuscript does a (deceptively) good job of simplifying a very complicated system for a general audience, but have oversimplified their results in the case of expert readers. As I will argue below, there are serious omissions in the submitted manuscript which bear upon the robustness of the observed magnetic textures and the physics that underlie the SHP images obtained by the authors. Hopefully, many of these technical issues can be adequately covered by additions to the Supplementary Material (SM), while preserving a short paper worthy of publication in Nature Communications.

Response:

We would like to thank the referee for his/her detailed report, which we found very helpful to improve our manuscript. In the following, we address all the comments of the referee and indicate the changes made in the revised manuscript.

I. Problems with Spin Ice Terminology

A. Artificial spin ice (ASI) has been a subject of interest since the early work of Schiffer’s Group in 2006, and a corresponding set of technical terms has emerged over the years. However, these terms are currently evolving in response to a recent increase in the number and types of artificial (mesoscale) materials fabricated to exhibit frustration (see the Physics Today article by Ian Gilbert et al. in 2016). This has led to a generalization of ASI terms that is not yet settled.

In the context of frustrated metamaterials, the term, “ice” originally referred to frustration indicated by two possible lengths of oxygen-hydrogen bonds between tetrahedrally coordinated oxygen atoms, and the large entropy observed for cubic water ice at low temperatures. An analogous case of frustrated magnetism (including large low temperature entropy) was subsequently observed in tetrahedrally coordinated rare earth spins in pyrochlore structures by Ramirez and coworkers; here, degenerate configurations where two of four rare earth spins pointed into a tetrahedron, and two pointed outward, led to their designation as “spin ices”. This idea was further extended to “artificial spin ices” (ASI) by Schiffer and coworkers in 2006: They fabricated square ASI composed of

square arrays of elongated ferromagnetic dots of mesoscopic dimensions whose shape anisotropy caused them to behave as Ising dipoles. Most relevant, these classical Ising spins could be directly imaged by MFM and PEEM, which showed they obeyed a “two-in/two-out, Pauling ice rule” that governs the lowest energy spin arrangements of the square ASI (see the SM Fig. s1).

The term “magnetic charge ice” conventionally refers to regions of space (occupied by ferromagnetic nanostructures) in which the divergence of the magnetization is non-zero, as in the case of either of the two ends of elongated segments of uniformly magnetized ferromagnetic films (e.g., permalloy). Consider a Gaussian “pill box” with one end inside the ferromagnetic dot and the opposite end outside the ferromagnetic dot, which yields a nonzero magnetic charge density $\rho = -\text{div}M$ at both end surfaces of the dot. The corresponding magnetic charge is found from Gauss’ theorem: $Q = M \cdot n \, dA = MA$ (in terms of the surface normal n , the uniform magnetization M , and the end area A of the dot). Note that this magnetic charge is defined as the source charge for the magnetic field intensity H , not the magnetic induction B (which retains its fundamental zero divergence). The idea of magnetic charge is very important in ASI, since it leads to highly simplified Coulomb (“dumbbell”) models for quantitative estimates of magnetic energies and degeneracies (entropies) of various arrangements of Ising moments.

Response:

We thank the referee for his/her detailed clarifying remarks on the spin ice and magnetic charge terminology. We have thoroughly taken them into account when preparing the revised version of our manuscript..

In the present case, the authors state at the bottom of page 1 of the manuscript that they “design a new artificial ice state by using a nanostructured superconductor which introduces better controllability of magnetic charge order” (my italics). The nature of this “charge order” is introduced in Figure 1. In particular, Fig. 1a is intended to show how “magnetic charge” is defined in this paper.

An A side: It would help the reader if the signs of the fields and “charges” were defined in terms of a z - axis. I assume that the figures depict the x - y plane, and the undefined z -axis points into the paper.

Response:

Indeed, the panels in Fig.1 depict the x - y plane, while the signs of the fields and “charges” are determined by the z component of the local magnetic field. To make this point clear, we have indicated the coordinate axes in the figure.

Figure 1a tries to define a “positive” magnetic charge as indicated by a yellow blob, and a “negative” magnetic charge indicated by a blue blob, in order to correspond directly with the SHPM data presented in the paper. The reader gradually becomes aware that the colored blobs refer to the magnetic field intensity generated by the supercurrent density J that is compelled to flow around the antidot, and that Ampere’s law is applied to define the yellow or blue blobs that represent “magnetic charge”. This is where the paper starts to lose clarity, because the magnetic induction generated within the antidot volume by the supercurrent obeys $\text{div}B = 0$. Moreover, the magnetization $4\pi M = B - H = 0$ inside the empty antidot, so we must have $\text{div}H = 0$ inside the antidot, which demands that the magnetic charge is zero throughout the antidot. On the other hand, the SHP measures a non-zero magnetic induction B directed along the z -direction, as shown in Figs. 1e or 1f, which agrees with Ampere’s law. I conclude that the yellow and blue blobs cannot represent “magnetic charges” as they are defined in the current ASI literature.

Response:

We agree with the referee that the magnetic charge as well as the magnetization inside the antidot is zero, and the observed magnetic dipoles, strictly speaking, are not equivalent to dipoles formed by “magnetic charges” defined in the artificial spin ice literature. At the same time, the magnetic field patterns, measured for our magnetic dipoles, and those for magnetic charges, can be remarkably similar to each other.

In the case of our work, the measurement is performed in a lift-off mode: the Hall probe of our microscope is aligned at $\sim 1 \mu\text{m}$ away from the superconductor surface. As a result, the measured field patterns correspond to a relatively large distance from antidots, where the magnetic field at each end of an antidot has a distribution similar to that of a magnetic charge. We agree that, even in this case, it can be misleading to identify the yellow and blue blobs in the measured field distributions with the magnetic charges defined in the ASI literature.

Since the yellow/blue blobs always appear as a pair, we call them magnetic dipoles. In the revised manuscript, when mentioning each pole, we have replaced “magnetic charge” with “magnetic pole”.

B. If one asks, “where is the divergent magnetization?”, one is reminded that that picture is best suited for a magnetostatics problem, not for a case of a variable source current J . This leads to another set of questions concerning the detailed nature of $J(r)$, which is not well-enough defined/discussed in the paper, given that it is a source of the blue and yellow blobs seen in SHP.

The blobs shown in various figures evidently represent z -components of field generated by two source currents, one due to Meissner “edge currents” (see Fig. 1a), and one due to trapped/pinned “vortices” (authors’ terms) in the antidots via field cooling (see Fig. 1d).

These two contributions need to be better explained in the early part of the paper and/or in the Supplementary Material (SM).

Response:

The Meissner current is induced by applying an external magnetic field. For a superconducting stripe, the in-plane Meissner current can be expressed as

$$J(x) = -2Hx(w^2 - x^2)^{-1/2}, \quad (1)$$

where w is the half width of the superconducting stripe, and x is the distance from its axis. The Meissner current flows along the edges of the sample.

For a free vortex, the distribution of its supercurrent is circularly symmetric with respect to the normal-state core. At large distances from the vortex core, the current density on the surface of the superconductor follows the $1/r^2$ law which has a simple expression [1]: $J_s(r, 0) = \Phi_0 [\coth(d/\lambda) + \operatorname{csch}(d/\lambda)] / 2\pi\mu_0\lambda r^2$, where r is the distance to the center of the vortex core, Φ_0 is the flux quantum, d is the sample thickness, λ is the penetration depth. For a vortex trapped by an elongated antidot with sizes larger than the vortex core, the supercurrent distribution is significantly modified: the current lines “expelled” from the antidot form two pronounced density maxima near the opposite ends of its long axis.

Since an antidot is much smaller than its distance from the nearest edge of the sample, the Meissner current density, given by Eq. (1), remains nearly constant if variations of this distance are of the order of the antidot size. Flowing towards an elongated antidot, the Meissner current is redistributed to form two half-circular current flows. If the angle between the initial Meissner current and the long axis of the antidote is sufficiently large, the redistributed Meissner current density takes maximum values at the opposite ends of the antidot long axis – resembling, in a sense, the distribution of the trapped-vortex supercurrents. However, while at one end of the antidot, the Meissner current is added to the vortex supercurrent, enhancing the local magnetic field, at the other end of the antidot, the Meissner current and the vortex current tend to cancel each other so that the local field is suppressed.

Following the referee’s advice, we have added such explanation of the two current sources in the supplementary material.

What I view is an incorrect use of the term, “magnetic charge” is compounded many times over in the discussion at the bottom of page 2 of the manuscript. There is also no discussion or clarification of the fact that the total magnetic flux must be quantized in the region of each antidot, according to the arguments originally given by London (see Introduction to Superconductivity, by M. Tinkham, page 122). The London picture demands that a close coupling (constraint) exists between the local supercurrent density $J(r)$ and the number of quantized fluxoids trapped by a given antidot. The proximity of the supercurrent flow to any antidot shown in Fig. 1 complicates derivation of an analytic expression for

either $J(r)$ (the source of the superconducting magnetization of interest, and therefore the “magnetic charge” present in the same region of space) or the spatial extent of the magnetic field intensity inside/near a given antidot.

Response:

In a superconducting condensate, the fluxoid quantization follows from the phase coherence of the macroscopic wave function along a closed path (e. g. a contour encircling one or more vortices). Strictly speaking, this assumes that the path never has to go through a region with zero order parameter (e.g., see the solid circles in Fig. r1a). For the contours, shown by dashed lines in Fig. r1b, the aforementioned assumption is violated. This violation seems sufficient – at least, formally – to argue that the flux, corresponding to the pole (yellow) or antipole (blue) of the magnetic dipole, no longer must be quantized. At the same time, as implied by the results of [27], this flux will remain non-quantized even in the case of a blind antidot, where the corresponding fluxoid is well defined and equals 0 (unless the magnetic dipole transforms into a fully developed vortex-antivortex pair [27]). It should be emphasized that such a “classical” behavior of magnetic poles does not violate at all the fluxoid-quantization rule. Indeed, these “noninteger” magnetic poles and antipoles always appear in pairs as bound dipoles, so that the total flux/fluxoid corresponding to a dipole is zero and therefore remains quantized. The magnetic dipoles originate from a rather “classical” effect: local redistribution of currents in the vicinity of a defect. In a strong contrast to the processes of vortex penetration or vortex-antivortex pair generation, this redistribution does not change the topology of the superconducting condensate.

We have now added a short discussion in the manuscript to clarify this point.

Figure r1. Different types of closed contours in a superconducting state with a magnetic dipole. The rectangle indicates the antidot which is nonsuperconducting.

C. On page 1, third paragraph, the authors refer to the work of Kwok’s Argonne Group on “rewriteable artificial spin ice” which focuses on magnetic charge

order rather than Ising spin order in a modified ASI [27]. The authors state that “no spin ice rule is applicable here”. This statement should be briefly clarified, since this work is seminal to the present paper.

Response:

In ASI systems, each spin is represented by a ferromagnetic island. Under the spin ice rule, the ground state (type-I vertex) of a square ice requires the “spins” to follow two-in/two-out distribution, corresponding to the lowest energy state of the system. In Ref. [27], the authors realized the same magnetic charge distribution as Schiffer’s original square spin ice. However, the arrangement of spins is completely different. An important message of Ref. [27] is that one can generate the same distribution of magnetic charges from a totally different spin arrangement. There is no need to focus just on the spins. On this point, our current work follows the same strategy to generate a “magnetic field” distribution similar to that for the ground state of Schiffer’s original square spin ice. Following the referee’s advice, we have briefly clarified this point in the revised manuscript.

D. On page 1, the fourth paragraph states, “The magnetic dipoles that emulate the spins, are generated by applying a current around elongated antidots. Such a way allows switching on/off the magnetic dipoles by simply controlling the current. Also, the strength of the magnetic dipoles can be tuned by changing the current density...” I assume “magnetic dipoles” is a term referring to either a single, or two oppositely directed dipoles in Fig. 1. I suggest the authors realize the wisdom of their own wording here, and use some variation of “dipole” (instead of “magnetic charges”) to describe the CW and CCW source currents (Fig. 1a) that generate the yellow and blue blobs, respectively, that represent the magnetic field concentrations along the z-direction in Figs. 1 and 2. The true source currents $j(r)$ that generate the “dipoles” must flow within the x-y plane inside the superconductor, and are not located at the positions of the yellow and blue blobs inside an empty antidot.

Response:

We agree with the referee that the term “magnetic charge” defined in the artificial spin ice literature, is not quite appropriate to describe the magnetic dipoles observed in our present research. The main message of our work is that, by using nanostructured superconductors, we are able to design/generate/manipulate magnetic structures that mimic magnetic field patterns of various magnetic charge states in artificial spin ice systems. The advantage of using nanostructured superconductors is that we can simply switch on/off such magnetic patterns by controlling current density. By combining with vortex matter, we are also able to selectively control the number of magnetic poles.

In the revised manuscript, instead of exploiting the magnetic charge ice terminology, we use the term “magnetic pole” to describe the observed positive/negative magnetic field distributions. Also, in the revised manuscript, we put more emphasis the new approach in

designing different geometrical lattices of moments, including magnetic-charge-ice-like structures.

E. The authors remark in the second paragraph on page 2, “Instead of using ferromagnetic islands, in our sample, the magnetic charges are generated by applying a current that flows around antidots. As schematically shown in Fig. 1a, positive (yellow) and negative (blue) magnetic charges are formed at two ends of the elongated antidot [28, 29].” Given my arguments in D above, this usage of “magnetic charge” is misleading (and wrong).

One might try to emulate the field distribution of the yellow and blue blobs by imagining two rectangular current loops (dipoles) placed at opposite ends of the rectangular antidot such that their currents and magnetic fields cancel near the midpoint of the antidot. That is my crude suggestion for an analogue of the “dumbbell charge” model for ASI.

Response:

We have replaced the term “magnetic charge” with “magnetic pole” when describing the observed yellow/blue blobs in our manuscript.

Indeed, the observed magnetic dipole (yellow/blue blobs) at an antidot is similar to that generated by two current loops with clockwise and counterclockwise currents.

F. On page 2 of the manuscript, a “pinned vortex lattice” refers to Fig. 1d, which shows an SHP image of magnetic flux trapped by field cooling through T_C . I recommend the authors (optionally) adhere to the term “fluxoid” here and elsewhere to avoid confusion with the quantized “Abrikosov vortex” present in the Type II mixed state, or “multiquantum vortices” found in antidot lattices in Type I materials (see G.R. Berdiyrov, M.V. Milosevic, F.M. Peeters, *Physica C* 437–438 (2006) 25). The authors also refer to “recently predicted vortex ice” in the abstract, and “vortex ice” on page 2 of the manuscript, which is not adequately defined or referenced (it is only vaguely mentioned in the lead paragraph on page 1, with referencing [7-9]).

Response:

We would like to mention that our sample is a superconducting film in the type-II regime. In the reply to comment II-A below, we describe how the critical parameters and the Ginzburg-Landau parameter were estimated. We apologize for the confusion caused to the readers. Now we have clearly indicated the critical parameters in the Method part of the revised manuscript. Also in the supplementary material, we added the details which clarify how these parameters were determined. In the supplementary material, we have added a figure (Fig. s7) and a short description to introduce the term “vortex ice”.

II. Problems with Sample Characteristics

There are several ambiguities regarding the nature and behavior of the patterned Pb films used in this study. Some are probably trivial, requiring only brief clarification in either the main text or the SM; others bear on the robustness of the ice behavior and how it is controlled.

A. The samples consist of Pb(90nm)/Ge(10nm)/Au(35nm) heterostructures with unstated lateral dimensions (x-y plane). The superconducting transition temperature $T_C = 7.35$ K, which differs slightly from the accepted bulk Pb $T_C = 7.190$ K. The web values of the relevant bulk Pb parameters are coherence length $\xi_0 = 83$ nm, London penetration depth $\lambda_L = 37$ nm. We therefore expect an effective, thin-film penetration depth $\lambda_{eff} = \lambda_L(\xi_0/t)^{1/2}$, where t is the film thickness in cases where $t \ll \xi_0$, which indicates near-bulk behavior is expected for these samples, since $t = 90$ nm $> \xi_0$. These numbers also indicate that a corresponding unpatterned Pb film is a Type I superconductor, since the bulk Ginzburg-Landau $\kappa \approx \lambda_L/\xi_0 = 0.45 \ll 0.7$. This parameter check should be added to a sample description (so far missing) in the SM.

Response:

We have indicated the dimensions ($200 \times 200 \mu\text{m}^2$) of our sample in the Method part of the revised manuscript. Enhanced T_c in superconducting films have already been reported in many materials. For our Pb film, we have determined T_c with two different ways. First, by measuring the local ac susceptibility, we observe $T_c \sim 7.15$ K; second, by directly imaging one vortex with slowly increasing temperature. As shown in the Fig. ra below, the vortex disappear between 7.3 K and 7.4 K. Therefore, we determine the T_c of the our sample as 7.35 K. Compared with transport and magnetization measurements of a bulk sample, our method gives a more precise estimate of T_c , due to the fact that, close to T_c , the vortex matter is already in the liquid regime and shielding of external magnetic field is rather weak. We notice that, $T_c = 7.3 \pm 0.05$ K for Pb has also been reported by carefully design the experiment [S. Westerdale, published on MIT website (2010), see: <http://web.mit.edu/shawest/Public/jlab/Supercon/superpaper.pdf>].

Pure bulk Pb is a well known type-I superconductor with the critical parameters indicated by the referee. In our present research, the sample is a superconducting film which was prepared with e-beam evaporation technique. The base pressure of the system is 2×10^{-8} Torr, and the pressure did not exceed 10^{-7} Torr during the evaporation. The evaporation rate is 1 \AA/s . In order to ensure a uniform layer growth for the Pb, the substrate was cooled by using liquid nitrogen during the preparation.

Normally, the properties of superconducting films, prepared with the e-beam technique, strongly depend on the evaporation pressure, substrate temperature and so on. For example, the use of a cold (room temperature) substrate leads to formation of a dirty Nb film, while a heated substrate ($780 \text{ }^\circ\text{C}$) results in clean limit samples [PRB 72, 014515 (2005)]. Our Pb sample is prepared at 77 K. As a result, a dirty limit sample is obtained with a relatively large penetration depth and small coherence length. Therefore, the

Figure r2. (a) Magnetic field distribution of a vortex at various temperatures. With increasing temperature, the vortex disappears between 7.3 K and 7.4 K. (b) Field profile along the dashed line for the vortex at $T=6$ K. Solid line in (b) is the monopole model fit. (c) $\lambda+z_0$ vs $(1-t^4)^{-1/2}$. Solid line in (c) is the linear fit to the data.

parameters of pure bulk Pb cannot be directly used to estimate the critical values of our thin films.

We have estimated the penetration depth and coherence length of our sample. The penetration depth is determined by using the monopole model [PRB **88**, 174503 (2013)] to fit the vortex field profile measured at various temperatures. Figure r2a shows the SHPM images taken at various temperatures indicated above each image. By fitting the magnetic field profiles along the dashed line, like it is shown in Fig. r2b for $T=6$ K, we can get the value of $\lambda+z_0$ at different temperatures, where λ is the penetration depth and z_0 is the distance between the two-dimensional electron gas (TDEG) of our Hall cross and the sample surface (this distance is constant). According to the two-fluid model, the penetration depth follows a linear dependence on $(1-t^4)^{-1/2}$ with $t = T/T_c$. As shown in Fig. r2c, from the slope of the linear fitting line, we can get the penetration depth at zero temperature. From the fitting, we get $\lambda(0)=151$ nm.

To determine the coherence length, we measured the temperature dependence of the in-phase ac susceptibility at different magnetic fields. The temperature dependence of H_{c2} can be deduced from the χ' - T curve. The coherence length is calculated using the relation $H_{c2}(T)=\Phi/2\pi\xi(T)$. With the two-fluid model, we estimate $\xi(0)=52$ nm. Therefore, for the Ginzburg-Landau parameter we obtain the value $\kappa \approx 2.9$, which places our sample well in the type-II regime. Also, from the literature [PRL **30**, 603-606 (1973)], the critical thickness

above which the Pb film transits from type-II to type-I superconductor is around $d=300$ nm, that is significantly larger than the thickness of our sample.

Moreover, we did not observe any giant vortices or normal domains in the reference sample. All the evidences above suggest that our sample is a type-II superconductor. Following the referee's suggestion, we have added a description of the sample parameters in the supplementary material.

B. Note that the characteristic lengths given above are zero-temperature values; the effective penetration depth and coherence length increase quite rapidly as temperature approaches T_C from below. The SHP images shown in Fig. 2 were apparently taken at $T = 4.2$ K (Figs. 2g, 2h); information regarding measurement temperatures is insufficient in the paper, and needs to be added in several places, as variations of temperature could drastically alter the observed behaviors.

Response:

In our manuscript, all the SHPM images were measured at 4.2 K, which is also the base temperature of our scanning Hall microscope. We have added this information in all figure captions and the main text of the revised manuscript.

C. The authors briefly point out that their SHP imaging was performed near the sample edge, due to the spatial distribution of the Meissner screening current: "The scanned area is chosen close to the sample edge which is parallel to the dashed lines" (page 3, first paragraph). It is unfortunate that the authors have not supplied an accurate sketch of the entire sample film, showing the geometry of the antidot lattice and its distance from the sample edges; indeed, the magnitude and geometry of the screening Meissner currents generated by applied field variations are the drivers of the magnetic ice configurations shown in Figs. 1 and 2. The dimensions of the antidot lattice include an approximate separation between antidots of 2400 nm (see Fig. 1d), which, with the above parameter check, implies the supercurrent and local fields can exhibit strong spatial dependence (depending on temperature) over the dimensions shown in Fig. 1. The equation given for the spatial dependence of $J(x)$ in the Methods section lacks a reasonable sketch or graph, and says nothing about the effects of the antidot lattice on $J(r)$, which could be quite complicated. A thorough discussion of such information would be very helpful to the expert reader, and should be added to the SM.

Response:

In the current research, Meissner current is used as the source to generate the magnetic dipoles. As shown by the expression for $J(x)$, the Meissner current density reaches its

maximum close to the sample edges and decreases with increasing the distance from the edges. Our sample has a square geometry with the dimensions of $200 \times 200 \mu\text{m}^2$ (see Fig. r3). The inhomogeneity of the Meissner current results in an inhomogeneous magnetic dipole pattern across the sample. Close to the sample edges, the magnetic fields induced by the dipoles are the highest. In the manuscript, the scanned area ($16 \times 16 \mu\text{m}^2$) has been chosen close to the center of one sample edge (see blue square in the Fig. r3). Within the relatively small scanned area, variations in the magnetic dipole intensity (caused by the dependence of the Meissner current density on the distance from the sample edge) are negligible.

Figure r3. Optical image of the measured sample. The scanned area is shown by the blue square close to the center of one sample edge. The red dashed line indicates the flow of the Meissner current, which is induced by applying an external magnetic field.

D. The Leuven Group and the Antwerp Theory Group of Peeters have published extensive work on the behavior of Type I films patterned with antidot lattices, and verified the existence of “multiquantum vortices” and other interesting complications due to finite sample and antidot lattice dimensions, temperature, applied field, and so on, which are relevant to the present work.

(1) What kinds of restrictions (especially regarding temperature and field/current penetration) do these details put on the measurements of magnetic ice behavior reported in this paper?

Response:

As we have shown, our sample is a type-II superconductor. In the Meissner state, i.e, at external magnetic fields smaller than the penetration field, the magnetic patterns are only determined by the local Meissner current which is proportional to the external magnetic field. However, once the external magnetic field is bigger than the penetration field (dependent on temperature), and the Meissner current density reaches the critical value at the sample edges, vortices start to penetrate into the sample and occupy some antidots. As a result, the regular magnetic patterns will be destroyed. So the key issue to observe ice-like patterns in our research is to make sure that the sample remains in the Meissner state at the measurement temperature.

(2) How homogeneous is the sample magnetic texture across the entire sample?

Response:

The current source we used to generate the magnetic dipoles is the Meissner current. The magnetic textures strongly depends on the distribution of the Meissner current density. As shown in Fig. r3, the Meissner current flows along the border of the sample, and it changes direction at the sample corner. Close to the opposite edge, the current direction totally reversed. As a result, the generated magnetic dipoles also change their polarities at the antidots. To avoid any confusion, in our manuscript, all the SHPM images were measured at the same position in the sample (blue square shown in Fig. r2). To make this point clear to readers, we have added a short discussion in the supplementary material.

(3) As the authors suggest in their introduction, dynamics and poor equilibration have presented serious problems in studies of artificial spin ices; do they complicate the characterization of the equilibrium properties of the present materials? Is the magnetization map of the sample stable with time after increments in the applied DC field; that is, are there magnetic relaxation effects and/or hysteresis due to domain wall motion in the (expected) intermediate state, or flux creep in a (unexpected) Type II mixed state?

Response:

The observed magnetic patterns are stable and totally reversible with changing the DC magnetic field as long as the field value is smaller than the penetration field H_p . Our sample is a type-II superconductor. Once the external magnetic field is above H_p , vortices start to enter from the edges. In this case, the motion of vortices may be rather uncontrollable due to flux creep and avalanches. The magnetic field of vortices will overlap with the magnetic dipoles, leading to the formation of complicated patterns. In the present research, the applied external magnetic field never exceeds H_p . So no flux creep or avalanche need to be considered here.

E. On page 2, second paragraph, the authors state, "In this paper, instead of using transport current, a Meissner current, induced by applying an external field (smaller than the lower critical field H_{C1}), is used to generate the magnetic charges." On page 3, first

paragraph, “Figure 1d represents the vortex lattice observed at first matching field, where each antidot is occupied by one vortex. The scanned area is chosen close to the sample edge which is parallel to the dashed lines.” On page 3, second paragraph, we read, “the intensity of magnetic charges (i.e. the interaction strength) can be tuned by varying the Meissner current density (magnetic fields), which is directly proportional to the external magnetic field at field values smaller than the penetration field [28].”

This wording implies that the authors see a need to avoid intermediate state domain formation, or other complications involving the sample edges or interior. Why not use transport current to control the ice behavior? These issues should be at least acknowledged and better explained for the expert reader.

Response:

We agree with the referee that, by using a transport current to generate the magnetic patterns (for example, in the normal state, one can even generate such magnetic patterns at room temperature), a more uniform distribution of magnetic dipoles can be expected. However, one should bear in mind that, in the Meissner state, due to the shielding of magnetic field, the transport supercurrent tends to have maximum density only at the edges of the superconductor [Supercond. Sci. Technol. 15, 82-89 (2002),]. As a result, we would observe a similar magnetic pattern as that in the case of Meissner current. However, we have noticed a recently published literature (Scientific Reports 8,1716(2018)), where the authors studied the current distribution at high current densities. They have shown that the transport current has a non-uniform distribution at low values, while a crossover to a uniform distribution occurs at critical current. In our present research, due to the imperfection of the sample edges, it is rather difficult to control the current at such high values. The main idea of the current research is to introduce a new way to design geometric lattices of magnetic moments. The Meissner current seems quite suitable for this purpose. To make this point clear, we have added a paragraph in the supplementary material.

F. On page three, third paragraph, it is stated, “the switchable magnetic-charge-ordering state only depends on the direction and density of the applied current. For this reason, it is protected against distortions, caused, for instance, by the stray magnetic field generated by vortices.”

Are the authors making a comparison with the Type II mixed state with Abrikosov vortices here? Can possible interstitial “multiquantum vortices” in the Type I intermediate state be relevant here? How does flux quantization affect things? Please clarify this wording.

Response:

As we have replied before, our sample is a type-II superconductor. Only single quantum vortices exist in our sample at magnetic fields lower than the first matching field. At higher magnetic fields, an antidot can trap more than one vortex to form “multiquantum” vortices. However, those vortex configurations are beyond the focus of the current research.

When using a traditional artificial spin ice (ASI) as a substrate and coupling it with other systems, e.g., a superconducting layer, the magnetic moments of nanomagnets in the ASI layer will be affected by the stray magnetic fields of vortices. When such interaction is strong, the moments of nanomagnets may even flip. As a result, the substrate layer is modified. However, in our case, the magnetic dipoles are fully determined by the applied current. The field density and polarity don't change as long as the current is stable. This is a big advantage when using such magnetic patterns as a substrate to couple with other systems. We have briefly clarified this in the revised manuscript.

III. Other Remarks

1) A better discussion of “frustration” in the present context would be useful for the general reader: How do the authors define/quantify frustration in their new ice? This should be associated with a macroscopic (i.e., proportional to system size) residual entropy associated with degeneracies among dipole distributions shown in Fig. 4, for example. Figure 4 and associated text are poorly motivated in the current manuscript (just “tacked on” at the end of the paper). Why were Figs. 4g and 4h shown? This should help clarify the discussion of ice degeneracy for a non-expert.

Response:

The origin of frustration is twofold. Firstly, the elongated vortex cores make the vortex-vortex interaction anisotropic. Frustration in the vortex-vortex interaction arises from the elongation of vortex cores in the direction of the closest neighbor. As a result, attractive vortex-vortex interaction appears. Secondly, in each unit cell of Fig. 1c, there are three antidots which are arranged into a triangular lattice. Such arrangement introduces geometric frustration, which is similar to that in the kagome lattice as studied in Ref. [PRB 96, 024510 (2017)]. As a result, degenerate states with the same energy are formed. For example, the two configurations of type-I/2 (or the two configurations of type-II/2) in Fig. 4 have the same interaction energy. Our brief discussion of the results shown in Fig. 4 is just aimed to show that our design may also be used to study complex frustrations in recently predicted vortex ice systems. We agree that further work is needed to interpret in detail the intricate vortex patterns observed. However, this is beyond the focus of the current research.

2) I found the discussion of the proposed relationship of the authors' dipole ice with effects of “elongated cores” and the “anomalous matching effect” in nanostructured superconductors to be vague and therefore not very useful for readers. Additional explanations might add weight to these possibilities.

Response:

In type-II superconductors, vortices have circular cores. The interaction between two vortices is purely repulsive. Under such repulsive interaction, vortices form triangular lattice. However, for vortices with elongated cores, the interactions become anisotropic. Frustration in the vortex-vortex interaction arises from the elongation of vortex cores in the direction of the closest neighbor [*PRB* 95, 104519 (2017)]. This leads to the appearance of an attractive interaction between vortices. As a result, vortices tend to form stripes instead of triangular lattice. In our current work, the trapped vortices were forced to follow the elongated antidot geometry. As a result, frustrated interactions occur together with the geometric frustration. The two effects combined lead to the formation of intricate vortex patterns.

Anomalous matching effect was observed in an artificial vortex ice made with nanostructured superconductors. Normally, in superconductors with periodic lattice of antidots, the critical current has a maximum value in the absence of magnetic field, and reaches peak values at integer matching fields. However, it is found that, in artificial vortex ice systems with geometric frustration, the critical current density reaches its maximum at half matching field, even higher than the value observed at zero field [*PRL* 111, 067001 (2013)]. Such anomalous matching effect was attributed to the ground state distribution of vortices at half matching field.

In order to make these two points clearer for the readers, we have added a brief discussion in the revised manuscript.

3) Figure 1d: The white arrowheads are missing, and there is not color scale shown. 4) Figure 4g: The square cell line for Type I/1 is displaced.

Response:

We thank the referee for pointing out these shortcomings. Now we have corrected all these items.

REVIEWERS' COMMENTS:

Reviewer #1 (Remarks to the Author):

The authors have satisfactorily revised the manuscript to avoid possible confusion with artificial spin ice (which is mainly about the study of frustrated many-body interactions and mobile magnetic charges). It seems to me that this is an interesting and technically difficult experimental work. As the authors also discussed in the reply and the revised version that the magnetic-pole pattern created in their array could be coupled to other meta-materials, from which interesting physics and phenomena are expected.

The authors also carefully answered the comments/criticisms of the other referees. I recommend the publication of this paper.

Reviewer #2 (Remarks to the Author):

The authors have properly answered my previous concerns addressed specially to the superconducting and experimental aspects.

Reviewer #3 (Remarks to the Author):

The authors have made numerous corrections to the manuscript text and the supplemental information to accommodate my many criticisms of an earlier manuscript. Most importantly, the authors have clarified the similarities and differences of their system with typical artificial spin ice materials.

I agree with the authors (and some of the other referees) that this work represents an interesting extension of magnetic (dipole)frustration into patterned superconducting films, and is acceptable for publication in Nature Communications.